# Atg1 kinase in fission yeast is activated by Atg11-mediated dimerization and cis-autophosphorylation

Zhao-Qian Pan[1,2], Guang-Can Shao[2], Xiao-Man Liu[2], Quan Chen[2], Meng-Qiu Dong[2,3], Li-Lin Du[2,3]*

[1]College of Life Sciences, Beijing Normal University, Beijing, China; [2]National Institute of Biological Sciences, Beijing, China; [3]Tsinghua Institute of Multidisciplinary Biomedical Research, Tsinghua University, Beijing, China

**Abstract** Autophagy is a proteolytic pathway that is conserved from yeasts to mammals. Atg1 kinase is essential for autophagy, but how its activity is controlled remains insufficiently understood. Here, we show that, in the fission yeast *Schizosaccharomyces pombe,* Atg1 kinase activity requires Atg11, the ortholog of mammalian FIP200/RB1CC1, but does not require Atg13, Atg17, or Atg101. Remarkably, a 62 amino acid region of Atg11 is sufficient for the autophagy function of Atg11 and for supporting the Atg1 kinase activity. This region harbors an Atg1-binding domain and a homodimerization domain. Dimerizing Atg1 is the main role of Atg11, as it can be bypassed by artificially dimerizing Atg1. In an Atg1 dimer, only one Atg1 molecule needs to be catalytically active, suggesting that Atg1 activation can be achieved through cis-autophosphorylation. We propose that mediating Atg1 oligomerization and activation may be a conserved function of Atg11/FIP200 family proteins and cis-autophosphorylation may be a general mechanism of Atg1 activation.

**\*For correspondence:**
dulilin@nibs.ac.cn

**Competing interests:** The authors declare that no competing interests exist.

## Introduction

Macroautophagy (hereafter autophagy) transports cytoplasmic materials into vacuoles or lysosomes for degradation. Pioneering works using the budding yeast *Saccharomyces cerevisiae* have identified a large number of autophagy-related (Atg) proteins that play crucial roles in autophagy (*Xie and Klionsky, 2007*; *Mizushima et al., 2011*). Atg proteins act in a hierarchical manner and among those acting most upstream is Atg1 (ULK1 in mammals), a serine/threonine kinase conserved across eukaryotes (*Matsuura et al., 1997*; *Suzuki et al., 2007*; *Cheong et al., 2008*; *Noda and Fujioka, 2015*).

Atg1 kinase activity is controlled by other Atg proteins. In *S. cerevisiae*, two scaffold proteins Atg13 and Atg17 are required for activating Atg1 kinase during nitrogen starvation-induced non-selective bulk autophagy (*Kamada et al., 2000*; *Yeh et al., 2010*; *Yeh et al., 2011*). In this Atg1 activation mode, monomeric Atg13, which can directly bind Atg1, makes multivalent interactions with dimeric Atg17, and thereby generates supramolecular assemblies, allowing concentration-dependent Atg1 autophosphorylation and autoactivation (*Yeh et al., 2011*; *Fujioka et al., 2014*; *Yamamoto et al., 2016*). The cytoplasm-to-vacuole targeting (Cvt) pathway in *S. cerevisiae*, a selective autophagy pathway that operates under nutrient-rich conditions, employs a different Atg1 activation mechanism where Atg17 is dispensable but Atg13 and another scaffold protein Atg11 are required (*Kamada et al., 2000*; *Kim et al., 2001*). In this Atg1 activation mode, Atg11 tethers Atg1 to the multimeric cargo-receptor complex to promote Atg1 autophosphorylation and autoactivation, and Atg13 plays a more indirect role by facilitating the Atg1-Atg11 interaction on the vacuole membrane (*Kamber et al., 2015*; *Torggler et al., 2016*). For both Atg1 activation modes in *S. cerevisiae*, it has been assumed that concentration-dependent Atg1 autoactivation occurs through trans-

autophosphorylation (*Yeh et al., 2011*; *Torggler et al., 2016*), but experimental evidence supporting this assumption is lacking.

Atg11 is conserved throughout eukaryotes, with a signature domain in its C terminus, termed Atg11 domain (*Li et al., 2014*). Unlike the situation in *S. cerevisiae* where Atg11 is dispensable for nitrogen starvation-induced bulk autophagy, in mammals, the Atg11-domain-harboring ortholog of *S. cerevisiae* Atg11, called FIP200 or RB1CC1, is essential for bulk autophagy (*Hara et al., 2008*; *Ganley et al., 2009*). Similarly, Atg11 orthologs in *Drosophila melanogaster*, *Arabidopsis thaliana*, and two yeast species *Kluyveromyces marxianus* and *Schizosaccharomyces pombe* are also indispensable for bulk autophagy (*Kim et al., 2013*; *Sun et al., 2013*; *Li et al., 2014*; *Yamamoto et al., 2015*). Mammalian and Drosophila Atg11/FIP200 proteins have been shown to promote Atg1 kinase activity (*Ganley et al., 2009*; *Nagy et al., 2014*). However, it is unclear how Atg11/FIP200 proteins enhance Atg1 kinase activity during bulk autophagy.

Here, we show that in the fission yeast *S. pombe*, Atg11 but not Atg13 is required for the Atg1 kinase activity. Atg11 promotes Atg1 activation by dimerizing Atg1. Contrary to what has been widely assumed, we find that Atg1 activation can occur through cis-autophosphorylation of Atg1. These findings shed new light on the Atg1 activation mechanism and the molecular roles of Atg11/FIP200 proteins.

## Results

### Fission yeast Atg11 but neither Atg13 nor Atg17 is required for normal Atg1 kinase activity

To determine whether the kinase activity of *S. pombe* Atg1 is important for autophagy, we introduced D193A and T208A mutations individually into endogenously mCherry-tagged Atg1. These two mutations are respectively the equivalents of the kinase-inactivating D211A and T226A mutations in *S. cerevisiae* Atg1, with the former disrupting the chelation of a catalytic metal ion and the latter abolishing the activation loop phosphorylation site (*Matsuura et al., 1997*; *Kijanska et al., 2010*; *Yeh et al., 2010*). Using an mYFP-Atg8 processing assay, which monitors the autophagy-dependent release of free mYFP upon the processing of mYFP-Atg8 by vacuolar proteases, we found that both mutations abolished nitrogen starvation-induced autophagy (*Figure 1A*), suggesting that like in *S. cerevisiae*, the kinase activity of Atg1 is essential for autophagy in *S. pombe*.

To directly assess the kinase activity of Atg1, we adopted a non-radioactive in vitro kinase assay based on the ability of Atg1 to use ATP-γ-S to modify its substrates with a thiophosphate group (thioP), which upon a derivatization reaction, can be detected by immunoblotting (*Figure 1B*; *Lo and Hollingsworth, 2011*; *Kamber et al., 2015*). When endogenously mCherry-tagged Atg1 was immunopurified from cells grown under nutrient-rich conditions and subjected to this in vitro kinase assay, an immunoblotting band reactive to the anti-thioP antibody was observed at a position overlapping the top edge of the mCherry-antibody-reactive band (*Figure 1C*), suggesting that Atg1 underwent in vitro autophosphorylation. Such a band was not observed for D193A and T208A mutants, confirming that they are indeed kinase dead (*Figure 1C*).

To explore how Atg1 kinase activity is controlled in *S. pombe*, we performed an in vitro kinase assay on YFP-tagged Atg1 immunopurified from mutants lacking Atg11, Atg13, Atg17, or Atg101, which are proteins acting with Atg1 in the autophagy initiation step (*Mizushima, 2010*; *Sun et al., 2013*). In *S. cerevisiae*, Atg13 and Atg17 are required for Atg1 activation during bulk autophagy (*Kamada et al., 2000*; *Kabeya et al., 2005*; *Yeh et al., 2010*). Surprisingly, we found that in *S. pombe*, Atg1 from *atg13Δ*, *atg17Δ*, or *atg101Δ* mutant exhibited autophosphorylation activities similar to that of Atg1 from wild type, under both nutrient-rich and nitrogen-starved conditions (*Figure 1D* and *Figure 1—figure supplement 1A*). By contrast, the in vitro autophosphorylation activity of Atg1 from *atg11Δ* mutant was almost undetectable (*Figure 1D* and *Figure 1—figure supplement 1A*). In addition, Atg1 immunopurified from *atg11Δ* mutant migrated faster than Atg1 from wild type or the other three mutants. Treating immunopurified Atg1 with lambda protein phosphatase did not alter the migration of Atg1 from *atg11Δ* mutant, but rendered Atg1 from wild type migrating as fast as Atg1 from *atg11Δ* mutant (*Figure 1—figure supplement 1B*), suggesting that Atg1 is phosphorylated in vivo in wild type but not in *atg11Δ* mutant. Taken together, these results indicate that in *S. pombe*, the autophosphorylation activity of Atg1 requires Atg11.

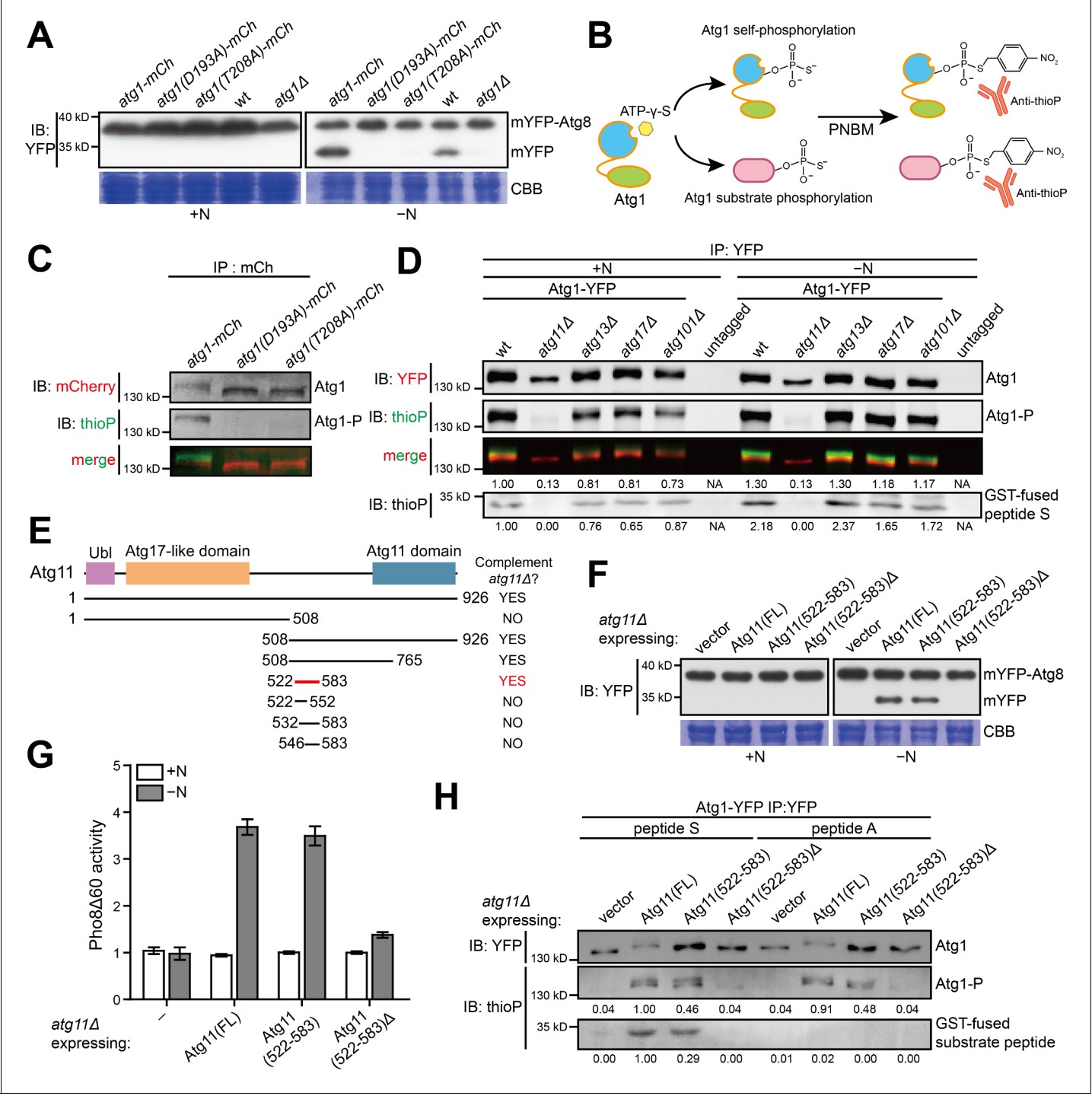

**Figure 1.** A 62 amino acid region in *S. pombe* Atg11 is necessary and sufficient for autophagy and for supporting the kinase activity of Atg1. (**A**) Mutations predicted to inactivate the kinase activity of *S. pombe* Atg1 abolished its autophagy function. The processing of endogenously mYFP-tagged Atg8 was used as a readout for autophagy. Cells were collected before (+N) and after shifting to a nitrogen-free medium for 12 hr (−N), and lysates were analyzed by immunoblotting using anti-YFP antibody. Coomassie brilliant blue (CBB) staining of PVDF membrane after immunodetection was used as a control for protein loading (*Welinder and Ekblad, 2011*). wt, wild type; mCh, mCherry; IB, immunoblotting. (**B**) Schematic of the non-radioactive in vitro Atg1 kinase assay. (**C**) In vitro Atg1 kinase assay confirmed that the kinase activity of *S. pombe* Atg1 was abolished by either D193A or T208A mutation. Endogenously mCherry-tagged Atg1 was immunopurified from cells growing in nutrient-rich medium, and subjected to the in vitro Atg1 kinase assay depicted in B. The immunoblotting signals were detected using the LI-COR Biosciences Odyssey infrared imaging system. IP: immunoprecipitation; Atg1-P, phosphorylated Atg1. (**D**) Atg11 but not Atg13, Atg17, or Atg101 is required for *S. pombe* Atg1 kinase activity in cells grown in a nutrient-rich medium (+N) and in cells shifted to a nitrogen-free medium for 1 hr (−N). Endogenously YFP-tagged Atg1 was immunopurified

*Figure 1 continued on next page*

Figure 1 continued

and analyzed by the in vitro kinase assay. Immunoblotting signal generated using anti-thioP antibody was quantified by densitometry and normalized to the immunoblotting signal of Atg1 protein, and reported as values relative to that of the wild-type sample under the +N condition. Results shown are the representative of three independent experiments. See *Figure 1—figure supplement 1A* for quantitation of data from three independent experiments. GST-fused peptide S is a non-self substrate (see *Figure 1—figure supplement 1C*). (E) Truncation analysis to identify the minimal region of Atg11 sufficient for its autophagy function. Conserved domains in *S. pombe* Atg11 are depicted at the top (see *Figure 1—figure supplement 2A* for domain conservation in Atg11/FIP200 proteins from representative eukaryotes). Starvation-induced vacuolar entry of mYFP-Atg8 was used as an autophagy readout to assess the ability of truncated Atg11 fragments to complement the autophagy defect of *atg11Δ* (see *Figure 1—figure supplement 2B* for imaging data). (F) Atg11(522-583) is necessary and sufficient for the autophagy function of Atg1. mYFP-Atg8 processing assay was performed as in A. FL, full length. (G) Pho8Δ60 assay confirmed that Atg11(522-583) is necessary and sufficient for the autophagy function of Atg11. *pho8Δ* cells expressing *S. cerevisiae* Pho8Δ60 were collected before (+N) and after nitrogen starvation for 4 hr (−N). The activity of Pho8Δ60 was normalized to the average activity of all samples under the +N condition. Data shown represent mean ± SEM (n = 3). (H) Atg11(522-583) is necessary and sufficient for the ability of Atg11 to support the ability of Atg1 to phosphorylate itself and the non-self substrate GST-fused peptide S. Endogenously YFP-tagged Atg1 was immunopurified and analyzed by the in vitro kinase assay. Results shown are the representative of three independent experiments. Quantitation was performed as in D. Values are relative to that of the reaction using Atg1 immunopurified from Atg11(FL)-expressing cells and using peptide S as the non-self substrate.

The online version of this article includes the following figure supplement(s) for figure 1:

**Figure supplement 1.** Atg11 is required for Atg1 kinase activity.
**Figure supplement 2.** Atg11(522-583) is sufficient for autophagy.

To examine whether Atg11 is required for the kinase activity of Atg1 toward substrates other than itself, we designed a substrate peptide (peptide S) based on the published consensus phosphorylation sites of *S. cerevisiae* Atg1 and human Atg1 ortholog ULK1 (*Papinski et al., 2014*; *Egan et al., 2015*; *Figure 1—figure supplement 1C*). For a negative control, we changed the only phosphorylatable residue in peptide S to alanine (peptide A). Both peptides were expressed and purified from *E. coli* as GST fusions (*Figure 1—figure supplement 1C*). Peptide S but not peptide A was phosphorylated by Atg1 immunopurified from wild type (*Figure 1—figure supplement 1D*). By contrast, Atg1 immunopurified from *atg11Δ* mutant failed to phosphorylate peptide S (*Figure 1D*, *Figure 1—figure supplement 1A*, and *Figure 1—figure supplement 1D*). Thus, Atg11 is required for Atg1 kinase activities toward not only itself but also a non-self substrate.

## Atg11(522-583) is necessary and sufficient for the autophagy function of Atg11 and for supporting Atg1 kinase activity

Atg11/FIP200 family proteins from diverse eukaryotic species including animals, fungi, amoebozoans, and plants all share three annotated domains: a ubiquitin-like (Ubl) domain (NCBI CDD cd17060), an Atg17-like domain (Pfam PF04108), and the hallmark Atg11 domain (Pfam PF10377; *Figure 1—figure supplement 2A*). To determine which region of *S. pombe* Atg11 is important for its autophagy function, we constructed a series of truncations and expressed them individually in *atg11Δ*. Remarkably, Atg11(522-583), a fragment only containing 62 residues and not overlapping with any of the three annotated domains, was able to rescue the defect of *atg11Δ* in starvation-induced vacuolar entry of mYFP-Atg8 (*Figure 1E* and *Figure 1—figure supplement 2B*). In the mYFP-Atg8 processing assay, Atg11(522-583)Δ, which lacks these 62 residues, failed to rescue *atg11Δ* (*Figure 1F*). By contrast, Atg11(522-583) rescued *atg11Δ* as well as full-length Atg11 (*Figure 1F*). Similar results were obtained using the Pho8Δ60 assay, where we expressed the autophagy reporter *S. cerevisiae* Pho8Δ60 in *S. pombe pho8Δ* background (*Yu et al., 2020*; *Figure 1G*). These results show that Atg11(522-583) is necessary and sufficient for the autophagy function of Atg11.

Next, we examined whether Atg11(522-583) can support the Atg1 kinase activity. Atg11(522-583), when expressed in *atg11Δ*, rescued the loss of the Atg1 kinase activity toward itself and peptide S, whereas Atg11(522-583)Δ failed to rescue (*Figure 1H*). Thus, Atg11(522-583) is necessary and sufficient for the ability of Atg11 to support the Atg1 kinase activity.

## Atg11(522-544) mediates a physical interaction between Atg11 and Atg1

The fact that only Atg11 but not Atg13, Atg17, and Atg101 is required for the kinase activity of *S. pombe* Atg1 suggests a particularly intimate relationship between Atg11 and Atg1 in fission yeast. Consistent with this idea, in an affinity purification coupled with mass spectrometry (AP-MS) analysis using Atg1 as bait, we were only able to detect Atg1 and Atg11 but not any other core Atg proteins (*Figure 2—figure supplement 1A*), suggesting that, under the conditions we used, Atg11 is the

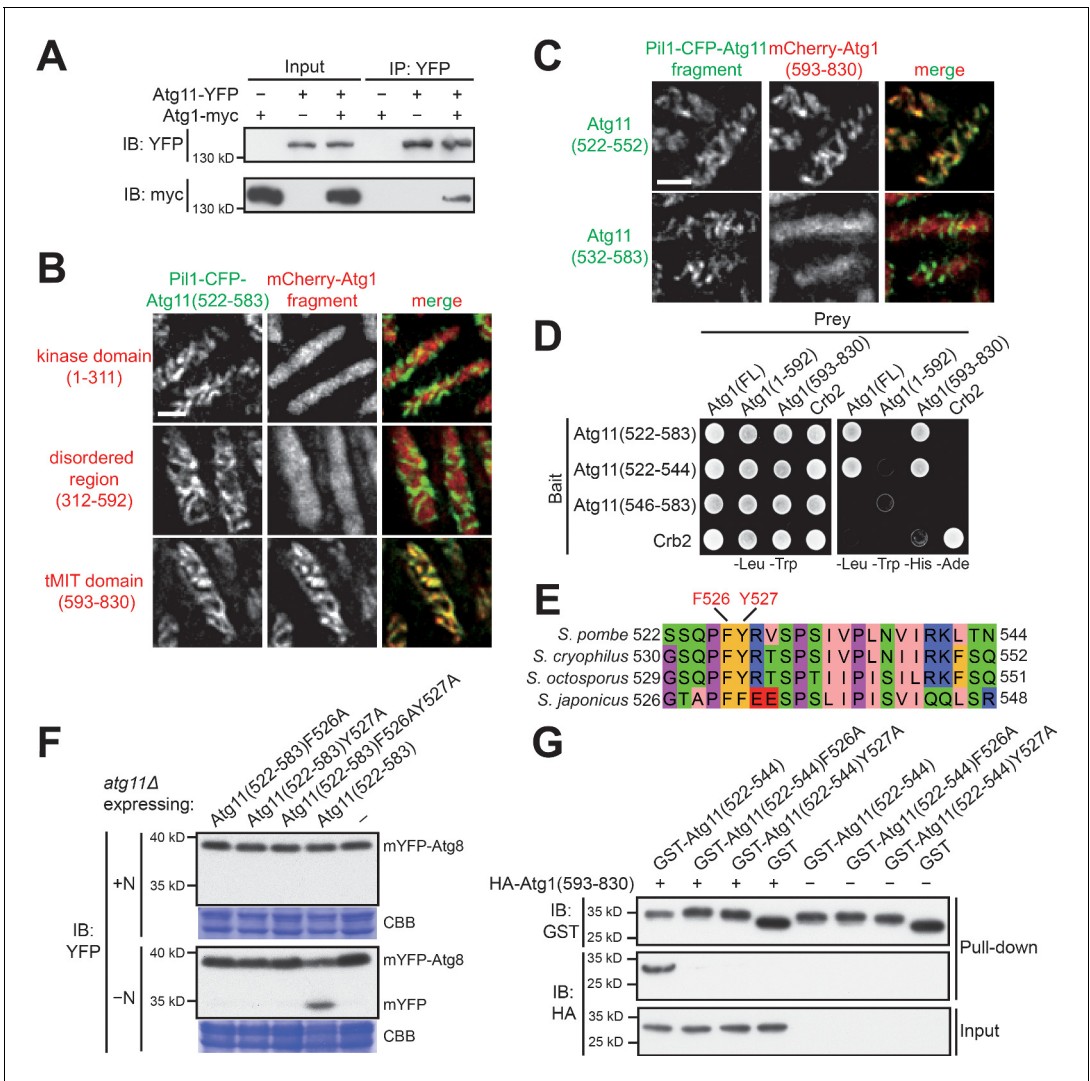

**Figure 2.** Atg11(522-544) mediates a specific and direct interaction with the tMIT domain of Atg1. (**A**) Coimmunoprecipitation assay showed that Atg11 physically interacts with Atg1. Atg11 and Atg1 were endogenously tagged with YFP and myc, respectively. (**B**) Pil1 co-tethering assay identified an interaction between Atg11(522-583) and the tMIT domain of Atg1. Log-phase cells co-expressing Pil1-CFP-Atg11(522-583) and an mCherry-tagged Atg1 fragment were examined by fluorescence microscopy. Scale bar, 3 μm. (**C**) Atg11(522-552) but not Atg11(532-583) interacted with the tMIT domain of Atg1 in the Pil1 co-tethering assay. Scale bar, 3 μm. (**D**) Y2H assay showed that Atg11(522-544) is sufficient for interacting with Atg1. Crb2, which can self-interact, served both as a positive control and a specificity control. (**E**) The sequence of *S. pombe* Atg11(522-544) was aligned to the corresponding regions of Atg11 proteins from three other fission yeast species. (**F**) Mutating either F526 or Y527 to alanine disrupted the autophagy function of Atg11 (522-583). mYFP-Atg8 processing assay was performed as in *Figure 1A*. (**G**) In vitro GST pull-down assay using recombinant proteins demonstrated a direct interaction between Atg11(522-544) and the tMIT domain of Atg1.

The online version of this article includes the following figure supplement(s) for figure 2:

**Figure supplement 1.** AP-MS analysis of Atg1-associated proteins and secondary structure prediction analysis on Atg11.

only strong physical interactor of Atg1 among core Atg proteins. We confirmed the Atg1-Atg11 interaction by coimmunoprecipitation using endogenously tagged proteins (*Figure 2A*).

To further dissect the Atg1-Atg11 interaction, we employed an imaging-based method that we have developed for examining protein-protein interactions, termed Pil1 co-tethering assay. Pil1 is a subunit of the plasma-membrane-associated eisosome complex, which forms visually distinctive filamentary structures in fission yeast (*Kabeche et al., 2011*). In the Pil1 co-tethering assay, we fuse a bait protein to Pil1. If another protein can interact with the Pil1-fused bait protein, it will exhibit co-localization with the bait on filamentary structures. Using this assay, we found that Atg11(522-583) interacted with the C-terminal tandem MIT (tMIT) domain of Atg1 but not with the N-terminal kinase domain or the disordered region situated between the kinase domain and the tMIT domain (*Figure 2B*). Upon further truncation, we found that Atg11(522-552) but not Atg11(532-583) interacted with the tMIT domain of Atg1 (*Figure 2C*). Thus, the N-terminal 10 amino acids of Atg11(522-583) are required for Atg1 binding.

Using the yeast two-hybrid (Y2H) assay, we narrowed down the Atg1-interacting region of Atg11 to Atg11(522-544) (*Figure 2D*). Secondary structure prediction showed that Atg11(522-544) may adopt a β-strand conformation between residues 526 and 529, and an α-helix conformation between residues 536 and 544 (*Figure 2—figure supplement 1B*). Based on an alignment of Atg11 from multiple fission yeast species, we identified two conserved aromatic residues in the β-strand section: F526 and Y527 (*Figure 2E*). mYFP-Atg8 processing assay showed that either F526 or Y527, when mutated to alanine, abolished the autophagy function of Atg11(522-583) (*Figure 2F*). Using recombinant proteins purified from *E. coli* to perform in vitro pull-down assays, we found that Atg11(522-544) directly interacted with the tMIT domain of Atg1 in a manner dependent on F526 and Y527 (*Figure 2G*). Together, these results show that Atg11(522-544) mediates a specific and direct interaction with Atg1, and this interaction is required for the autophagy function of Atg11(522-583).

## Atg11(546-583) mediates homodimerization

Atg11(522-552), a fragment sufficient for Atg1 binding, cannot support autophagy (*Figure 1E*), suggesting that Atg1 binding is not the only function of Atg11(522-583). The C-terminal part of Atg11 (522-583), starting from around residue 550, is predicted to adopt a coiled-coil conformation (*Figure 2—figure supplement 1C*). Because coiled coils often mediate self-interactions, we examined whether Atg11(522-583) and full-length Atg11 can self-interact. Using a coimmunoprecipitation assay, we found that both Atg11(522-583) and full-length Atg11 can self-interact (*Figure 3A* and *Figure 3—figure supplement 1A*). Using further truncated fragments to perform coimmunoprecipitation analysis, we found that Atg11(546-583) but not Atg11(522-552) was able to self-interact (*Figure 3B and C*). These results were corroborated by the Y2H assay (*Figure 3D*). Thus, the C-terminal part of Atg11(522-583) mediates self-interaction.

To determine the subunit stoichiometry of the Atg11(522-583) homo-oligomer, we purified Atg11 (522-583) from *E. coli* as a fusion with maltose-binding protein (MBP), a monomeric protein, and performed a gel-filtration analysis (*Figure 3E*). MBP-Atg11(522-583) eluted at an elution volume corresponding to that of a 101.8 kD protein. Because MBP-Atg11(522-583) has a calculated molecular weight of 50.6 kD, we concluded that Atg11(522-583) forms a homodimer.

To test the functional importance of Atg11 self-interaction, we removed amino acids 546–583 in full-length Atg11. This internal deletion disrupted the self-interaction of Atg11 (*Figure 3—figure supplement 1B*), abolished the autophagy function of Atg11 (*Figure 3—figure supplement 1C*), and abrogated the ability of Atg11 to support the Atg1 kinase activity (*Figure 3—figure supplement 1D*). Thus, Atg11(546-583)-mediated dimerization is essential for the normal functions of Atg11.

To determine whether mediating homodimerization is the only function of the C-terminal part of Atg11(522-583), we tested whether this part can be replaced by a heterologous dimerization domain. We fused a 32-amino-acid-long homodimerizing leucine zipper motif (LZ) (*Luo et al., 2015*) to Atg11 fragments and examined the ability of the fusion proteins to complement the autophagy phenotype of *atg11Δ*. LZ fusion fully restored the ability of Atg11(522-552) to complement *atg11Δ* (*Figure 3F and G*). Thus, dimerization is the main function of the C-terminal part of Atg11(522-583). Together, our results demonstrate that Atg11(522-583) is composed of two separable functional modules, an Atg1-binding domain and a homodimerization domain.

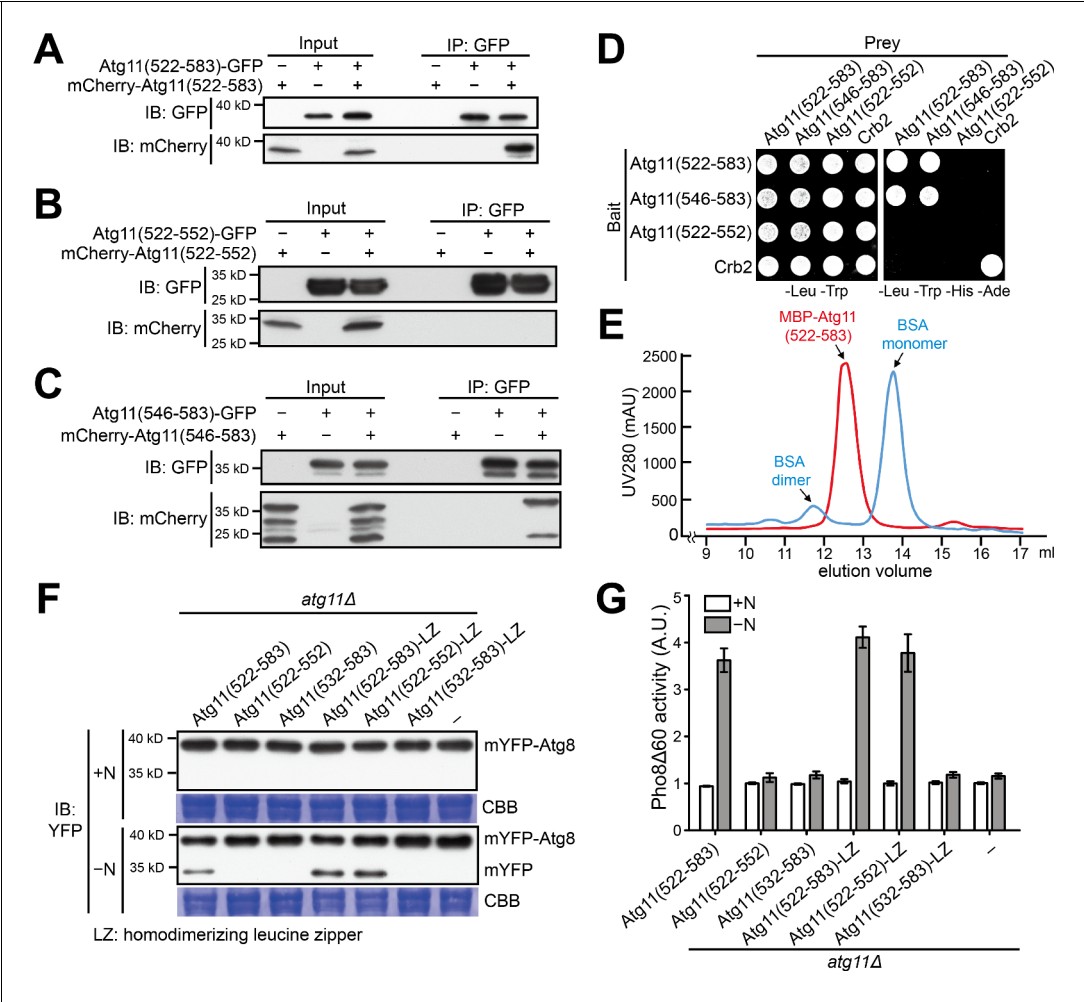

**Figure 3.** Atg11(546-583) mediates homodimerization. (**A–C**) Coimmunoprecipitation assay showed that Atg11(522-583) can self-interact (**A**), and Atg11 (546-583) but not Atg11(522-552) is sufficient for this self-interaction (**B and C**). To examine whether an Atg11 fragment can self-interact, its fusions with GFP and mCherry were co-expressed and the GFP-fused version was immunoprecipitated. (**D**) Y2H assay showed that Atg11(522-583) and Atg11(546-583) but not Atg11(522-552) can self-interact. (**E**) Atg11(522-583) forms a dimer. MBP-tagged Atg11(522-583) purified from *E. coli* was analyzed by gel filtration chromatography. Absorbance of eluates was monitored at 280 nm. Bovine serum albumin (BSA) monomer (66.5 kD) and dimer (133 kD) served as molecular weight markers. mAU, milli-absorbance units. (**F**) mYFP-Atg8 processing assay showed that fusing a heterologous dimerization motif (LZ) to Atg11(522-552) restored its ability to rescue the autophagy defect of *atg11Δ*. (**G**) Pho8Δ60 assay confirmed that Atg11(522-552)-LZ but not Atg11(522-552) can rescue the autophagy defect of *atg11Δ*. Data shown represent mean ± SEM (n = 3).

The online version of this article includes the following figure supplement(s) for figure 3:

**Figure supplement 1.** The self-interaction of Atg11 is important for the autophagy function of Atg11 and for supporting the kinase activity of Atg1.

**Figure supplement 2.** Atg11(522-583) and Atg11(522-552)-LZ but not Atg11(522-552) can support DTT-induced selective ER-phagy.

## Atg11(522-583) can support a selective autophagy pathway

In *S. cerevisiae*, Atg11 promotes selective autophagy by directly interacting with selective autophagy receptors (*Zientara-Rytter and Subramani, 2020*). The C-terminal region of *S. cerevisiae* Atg11, which encompasses the conserved Atg11 domain, is responsible for binding Atg19 and Atg32, the autophagy receptors for the Cvt pathway and the mitophagy pathway, respectively (*Yorimitsu and Klionsky, 2005*; *Aoki et al., 2011*). FIP200, the mammalian ortholog of Atg11, also uses its Atg11-domain-containing C-terminal region to bind autophagy receptors CCPG1 and p62 (*Smith et al., 2018*; *Turco et al., 2019*). As *S. pombe* Atg11(522-583) can support bulk autophagy but lacks the C-terminal region that harbors the Atg11 domain, we wonder whether it may lack the ability to support selective autophagy.

In *S. pombe*, ER stress-induced ER-phagy is the only known selective macroautophagy pathway (*Zhao et al., 2020*). We examined whether this selective autophagy pathway is functional by monitoring the vacuolar processing of an integral ER membrane protein Erg11 fused with CFP. When cells were treated with the ER stress inducer dithiothreitol (DTT), processing of Erg11-CFP occurred in a manner dependent on the ER-phagy receptor Epr1 (*Figure 3—figure supplement 2*; *Zhao et al., 2020*). DTT-induced processing of Erg11 was blocked in *atg11Δ* cells (*Figure 3—figure supplement 2*), indicating that, like bulk autophagy, ER stress-induced selective ER-phagy also requires Atg11. Interestingly, the ER-phagy defect of *atg11Δ* can be rescued by Atg11(522-583) (*Figure 3—figure supplement 2*). Thus, this 62 amino acid fragment of Atg11, even though lacking the region involved in autophagy receptor binding in other organisms, can support a selective autophagy pathway in *S. pombe*. Moreover, we found that Atg11(522-552)-LZ but not Atg11(522-552) can rescue the ER-phagy defect of *atg11Δ* (*Figure 3—figure supplement 2*), indicating that the dimerization ability of Atg11 is important for supporting selective autophagy.

## Artificial dimerization of Atg1 can bypass the requirement of Atg11 for autophagy and for Atg1 kinase activity

The findings that Atg11(522-583) possesses Atg1-binding and homodimerization abilities led us to hypothesize that the main role of Atg11 is to bring together two Atg1 molecules. This hypothesis predicts that we may be able to bypass Atg11 by artificially dimerizing Atg1. To test this prediction, we fused the homodimerizing LZ motif to the C-terminus of Atg1. This fusion protein, Atg1-LZ, but not wild-type Atg1, rescued the autophagy defect of *atg1Δ atg11Δ* double deletion mutant (*Figure 4A*). Furthermore, Atg1-LZ but not Atg1 immunopurified from *atg1Δ atg11Δ* cells exhibited robust in vitro autophosphorylation activity (*Figure 4B*). Thus, Atg11 is no longer needed when Atg1 is conferred an ability to homodimerize, supporting the idea that the main role of Atg11 is to dimerize Atg1, and thereby cause Atg1 activation.

## Dimerization of Atg1 can activate its kinase activity through cis-autophosphorylation

Dimerization-induced kinase activation can occur through either trans-autophosphorylation or cis-autophosphorylation of the activation loop (*Beenstock et al., 2016*). To determine which mechanism is used by Atg1, we employed the heterodimerizing protein pair GFP and GFP-binding protein (GBP) to drive artificial Atg1 dimerization in *S. pombe*. We found that, individually, both Atg1-GFP and Atg1-GBP can complement the autophagy defect of *atg1Δ* but not *atg1Δ atg11Δ* (*Figure 4—figure supplement 1A*), indicating that fusing GFP or GBP to Atg1 did not affect its function and either fusion alone cannot bypass Atg11. As expected, co-expression of Atg1-GFP and Atg1-GBP rescued the autophagy defect of *atg1Δ atg11Δ* (*Figure 4C* and *Figure 4—figure supplement 1B*). Interestingly, this rescue was not affected when the kinase-dead mutations D193A and T208A were introduced into Atg1-GBP or Atg1-GFP and was only abolished when both Atg1-GFP and Atg1-GBP harbored kinase-dead mutations (*Figure 4C* and *Figure 4—figure supplement 1B*). These results indicate that Atg1 activation can still occur when activation by trans-autophosphorylation is no longer possible (*Figure 4D*). Thus, we conclude that Atg1 can be activated by dimerization-induced cis-autophosphorylation (*Figure 4E*).

## Discussion

We show in this study that, in *S. pombe*, Atg1 kinase activity requires Atg11, thus explaining why Atg11 is essential for non-selective bulk autophagy in this organism. Unlike the situation in *S. cerevisiae* but similar to the situation in *S. pombe*, in mammals, *Drosophila melanogaster*, and *Arabidopsis thaliana*, Atg11/FIP200 family proteins are required for non-selective bulk autophagy, suggesting that in many eukaryotic species, Atg11/FIP200 may activate Atg1 independently of selective autophagy receptors and cargos. Lending support to this idea is the fact that animals and plants have Atg11/FIP200 family proteins but do not have Atg17-like proteins (*Li et al., 2014*). Thus, Atg11-mediated Atg1 activation may be a more ancient and more universal mechanism than the Atg17-dependent Atg1 activation mechanism used by *S. cerevisiae* during nitrogen starvation-induced bulk autophagy.

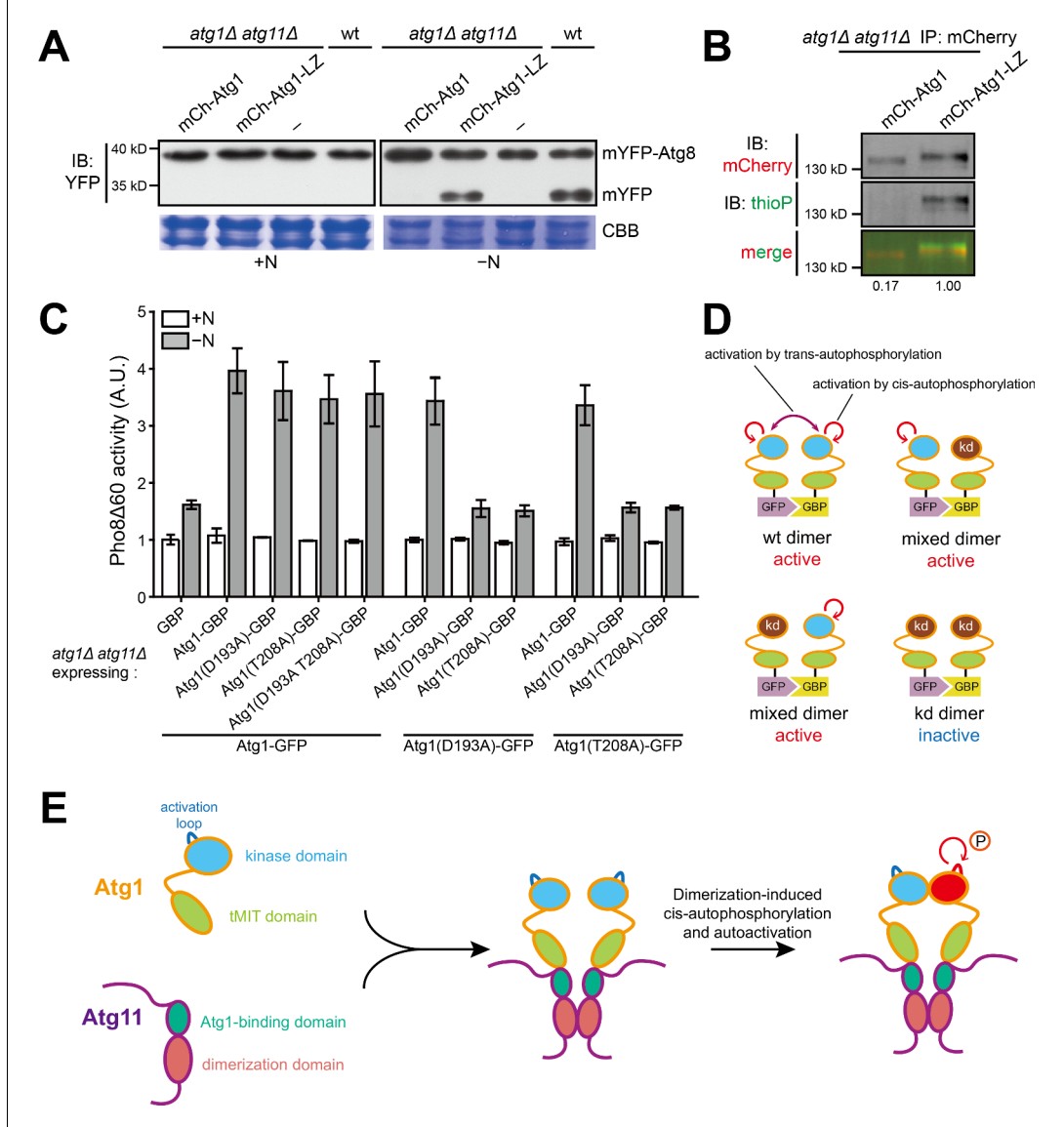

**Figure 4.** Dimerization of Atg1 activates its kinase activity through cis-autophosphorylation. (**A**) mYFP-Atg8 processing assay showed that fusing a heterologous dimerization motif (LZ) to Atg1 bypassed the requirement of Atg11 for autophagy. (**B**) In vitro Atg1 kinase assay showed that LZ-mediated dimerization of Atg1 bypassed the requirement of Atg11 for Atg1 kinase activity. Quantification was performed as in *Figure 1D*, and reported values are relative to that of the Atg1-LZ sample. Results shown are the representative of three independent experiments. (**C**) Pho8Δ60 assay showed that artificial heterodimerization of Atg1 mediated by GFP and GFP-binding protein (GBP) resulted in the bypass of Atg11 even when one Atg1 molecule in a dimer is inactive. GFP-tagged Atg1 was co-expressed with GBP-tagged Atg1 in *atg1Δ atg11Δ* cells. Data shown represent mean ± SEM (n = 3). (**D**) Diagram explaining the results in C. kd, kinase-dead. (**E**) Model depicting how Atg11 promotes Atg1 activation through dimerization-induced cis-autophosphorylation of Atg1.

The online version of this article includes the following figure supplement(s) for figure 4:

**Figure supplement 1.** Atg1-GBP or Atg1-GFP alone rescued *atg1Δ* but not *atg1Δ atg11Δ*, and their combination rescued *atg1Δ atg11Δ* even when one of them was kinase-dead.

Based on the molecular mechanisms revealed in this study, for Atg11 to act as an Atg1 activator in bulk autophagy, it needs to have two characteristics: Atg1-interacting ability and self-interacting ability. Both of these capabilities appear to be conserved throughout evolution. Atg11/FIP200 proteins in diverse organisms can physically interact with Atg1 (*Noda and Fujioka, 2015*; *Papinski and Kraft, 2016*). Self-interactions of Atg11/FIP200 proteins have been observed in *S. cerevisiae* (*Kim et al., 2001*; *Yorimitsu and Klionsky, 2005*), *C. elegans* (*Lin et al., 2013*), *Arabidopsis*

(*Li et al., 2014*), and mammals (*Turco et al., 2019*; *Shi et al., 2020*). In *S. cerevisiae*, even though Atg11 is dispensable for nitrogen starvation-induced bulk autophagy (*Kim et al., 2001*), it is required for bulk autophagy under carbon-depletion and phosphate-depletion conditions (*Adachi et al., 2017*; *Yokota et al., 2017*), suggesting that *S. cerevisiae* Atg11 may also promote Atg1 activation independently of selective autophagy cargos and receptors under certain conditions.

Dimerization-induced cis-autophosphorylation has been shown to be the activation mechanism for a small but increasing number of eukaryotic protein kinases including RAF, PKR, and RIPK3 (*Hu et al., 2013*; *Dey et al., 2014*; *Raju et al., 2018*). This type of activation mechanism is believed to be a multi-step process (*Beenstock et al., 2016*): first, the kinase domains of two monomers make a contact; second, this contact allosterically causes one or both monomers to adopt a prone-to-autophosphorylate conformation; third, cis-autophosphorylation of the activation loop happens and converts the kinase domain to an active conformation. Such a model may also apply to Atg1, whose kinase domain may only have a weak ability to transiently self-interact and thus requires Atg11 to promote kinase domain self-interaction at physiological Atg1 concentrations (*Figure 4E*). Further studies will be needed to reveal more mechanistic details of dimerization-induced Atg1 cis-autophosphorylation and determine to what extent the Atg1 activation mechanism we uncover here in fission yeast is conserved in other eukaryotic organisms.

# Materials and methods

## Key resources table

| Reagent type (species) or resource | Designation | Source or reference | Identifiers | Additional information |
|---|---|---|---|---|
| Gene (*Schizosaccharomyces pombe*) | *atg1* | PomBase | SPCC63.08C | |
| Gene (*Schizosaccharomyces pombe*) | *atg11* | PomBase | SPAC7D4.04 | |
| Genetic reagent (*Schizosaccharomyces pombe*) | Fission yeast strains used in this study | This paper | | See *Supplementary file 1* |
| Antibody | Anti-GFP (mouse monoclonal) | Roche | Cat# 11814460001; RRID:AB_390913 | 1:3000 dilution |
| Antibody | Anti-mCherry (mouse monoclonal) | Huaxingbio | Cat# HX1810 | 1:1000 dilution |
| Antibody | Anti-thiophosphate ester (rabbit monoclonal) | Abcam | Cat# ab92570; RRID:AB_10562142 | 1:5000 dilution |
| Antibody | Anti-myc (mouse monoclonal) | Abmart | Cat# M200002L | 1:3000 dilution |
| Antibody | Anti-GST (mouse monoclonal) | Abmart | Cat# M200007 | 1:3000 dilution |
| Antibody | Anti-HA (mouse monoclonal) | MBL | Cat# M180-3; RRID:AB_10951811 | 1:3000 dilution |
| Recombinant DNA reagent | Plasmids used for this study | This paper | | See *Supplementary file 1* |
| Chemical compound, drug | ATPγS | Sigma-Aldrich | Cat# A1388 | |

*Continued on next page*

*Continued*

| Reagent type (species) or resource | Designation | Source or reference | Identifiers | Additional information |
|---|---|---|---|---|
| Chemical compound, drug | p-Nitrobenzyl mesylate (PNBM) | Abcam | Cat# ab138910 | |
| Commercial assay or kit | GFP-Trap agarose beads | ChromoTek | Cat# gta-20; RRID:AB_2631357 | |
| Commercial assay or kit | RFP-Trap agarose beads | ChromoTek | Cat# rta-20; RRID:AB_2631362 | |
| Commercial assay or kit | Glutathione Sepharose 4 Fast Flow | GE Healthcare | Cat# 17-0756-01 | |
| Commercial assay or kit | Ni-NTA Agarose | QIAGEN | Cat# 30210 | |

## Fission yeast strain and plasmid construction

Fission yeast strains and plasmids used in this study are listed in *Supplementary file 1*. Genetic methods and composition of media are as described (*Forsburg and Rhind, 2006*). The deletion strains used in this study were constructed by PCR amplifying the deletion cassettes in the Bioneer deletion strains and transforming our lab strains. Strains expressing Atg1 fused with the YFP-FLAG-His6 (YFH) tag under native promoter was constructed by an overlap-extension PCR approach, using the Yoshida ORFeome plasmids as template (*Matsuyama et al., 2006*; *Yu et al., 2013*). Strains expressing Atg proteins fused with mCherry and YFP tags under native promoters were made by PCR-based tagging. Strains expressing Atg1 kinase-dead mutants fused with mCherry tag under native promoters were made using a CRISPR-Cas9 system (*Zhang et al., 2018*). Plasmids expressing Atg1 and Atg11 under the control of the *Pnmt1*, *P41nmt1*, or *P81nmt1* promoter were constructed using modified pDUAL vectors (*Wei et al., 2014*). The resulting plasmids were linearized with NotI digestion and integrated at the *leu1* locus, or linearized with MluI digestion and integrated at the *ars1* locus. Atg11 fragments were expressed using promoters of different strength to ensure that any fragment without autophagy function was expressed at a similar or higher level than the functional Atg11 protein used as the positive control. Strain expressing mYFP-Atg8 under the *atg8* promoter was as described (*Liu et al., 2018*). For the construction of GFP-GBP binding assay strains, an integrating plasmid expressing mCherry-Atg8 under the control of *P41nmt1* promoter was constructed by inserting sequences encoding *P41nmt1* promoter and mCherry-Atg8 into the integrating vector pPHA2H (pDB733; *Li et al., 2019*). The resulting plasmid was linearized with NotI digestion and integrated at the *pha2* locus.

## Non-radioactive Atg1 kinase assay

About 100 $OD_{600}$ units of cells were harvested and washed once with water. Cells were resuspended in buffer A (10 mM Tris HCl, pH 7.5, 150 mM NaCl, 0.5 mM EDTA, 0.5% NP-40) containing 2 mM PMSF and incubated at room temperature for 20 min. Cells were collected and resuspended in RIPA buffer (Sigma Aldrich) containing 1 mM PMSF (Sigma), 1 × protease inhibitor cocktail (Roche), 1 × phosphatase inhibitor PhosSTOP (Roche), and 0.05% NP-40, and mixed with 0.5-mm-diameter glass beads. Beads beating lysis was performed using a FastPrep-24 instrument (MP Biomedicals). The cell lysate was cleared by centrifugation at 13,000 rpm for 30 min. The supernatant was incubated with GFP-Trap beads (ChromoTek) or RFP-Trap beads (ChromoTek). After incubation, beads were washed three times with buffer A, and three times with kinase buffer (50 mM Tris HCl, pH 7.5, 200 mM NaCl, 10 mM $MgCl_2$, 0.05% NP-40). Beads were resuspended in 22 µL of kinase buffer and 1 µL of the consensus site Atg1 substrate (1 µg/ µL) was added. The kinase reaction was started by adding 1 µL of 10 mM ATPγS (Sigma). After incubating at 30°C for 20 min, the kinase reaction was quenched by the addition of 0.5 M EDTA to a final concentration twice that of the magnesium concentration. The thiophosphate group (thioP) was alkylated by adding 1.3 µL of 50 mM PNMB (Abcam) and incubating at room temperature for 2 hr. The alkylation reaction was stopped by boiling in SDS-PAGE loading buffer.

## mYFP-Atg8 processing assay

About 5 $OD_{600}$ units of cells were treated with 1.85 M NaOH and 7.5% (v/v) β-mercaptoethanol followed by 55% (w/v) trichloroacetic acid (TCA, Sigma; *Ulrich and Davies, 2009*). Samples were separated on 10% SDS-PAGE gels and then immunoblotted with an anti-GFP mouse monoclonal antibody (Roche), which can also recognize YFP.

## Fluorescence microscopy

Live-cell imaging was performed using a DeltaVision PersonalDV system (Applied Precision) equipped with an mCherry/GFP/CFP filter set (Chroma 89006 set) and a Photometrics EMCCD camera. Images were acquired with a 100×, 1.4-NA objective, and analyzed by the SoftWoRx software.

## Pil1 co-tethering assay

To examine a pair-wise protein-protein interaction, a bait protein was fused to Pil1-CFP, and a prey protein was fused to mCherry. Cells co-expressing both proteins were grown to log phase for fluorescence microscopy. To image the plasma-membrane associated filament-like structures formed by a Pil1-fused protein and its interactor, we acquired 8–10 optical Z-sections to ascertain the capture of a Z-section in which either the top or bottom plasma membrane is in focus. Images were processed using deconvolution in the SoftWoRx software.

## Pho8Δ60 activity assay

Pho8Δ60 activity assay was performed as described in *S. cerevisiae* (*Noda et al., 1995*) with some modifications (*Yu et al., 2020*). About 5 $OD_{600}$ units of cells were collected and washed with 0.85% NaCl, and then resuspended in 200 µL of lysis buffer (20 mM PIPES, pH 6.8, 50 mM KCl, 100 mM KOAc, 10 mM $MgSO_4$, 10 µM $ZnSO_4$, 0.5% Triton X-100, 2 mM PMSF) and incubated at room temperature for 20 min. PMSF was added to the final concentration of 4 mM and 0.5-mm-diameter glass beads were added. Beads beating lysis was performed using a FastPrep-24 instrument. The cell lysate was cleared by centrifugation at 13,000 rpm for 30 min. About 50 µL of the supernatant was added to 400 µL of reaction buffer (250 mM Tris-HCl, pH 8.5, 10 mM $MgSO_4$, 10 µM $ZnSO_4$, 0.4% Triton X-100, 5.5 mM 1-naphthyl phosphate disodium salt) to start the reaction. After incubation at 30°C for 20 min, 500 µL of 1 M glycine-NaOH (pH 11.0) was added to stop the reaction. Fluorescence was measured at 345 nm excitation and 472 nm emission using an EnSpire multimode plate reader (PerkinElmer). Protein concentration was determined by the BCA method (Thermo).

## Immunoprecipitation

About 100 $OD_{600}$ units of cells were harvested and washed once with cold water. Cells were mixed with 100 µL of lysis buffer (50 mM HEPES-NaOH, pH 7.5, 150 mM NaCl, 1 mM EDTA, 1 mM DTT, 1 mM PMSF, 0.05% NP-40, 10% glycerol, 1 × protease inhibitor cocktail) and 750 µL of 0.5-mm-diameter glass beads. Beads beating lysis was performed using a FastPrep-24 instrument. Then another 300 µL of lysis buffer was added and the cell lysate was cleared by centrifugation at 13,000 rpm for 30 min. The supernatant was incubated with GFP-Trap agarose beads. After incubation, the agarose beads were washed three times with lysis buffer. For immunoblotting analysis, bead-bound proteins were eluted by boiling in SDS loading buffer.

## AP-MS analysis

About 1000~1500 $OD_{600}$ units of cells were harvested, and the lysates were prepared using the bead-beating lysis method described in the Immunoprecipitation section above. The supernatant was incubated with GFP-Trap agarose beads for 3 hr. After incubation, the beads were washed twice with lysis buffer and twice with lysis buffer without NP-40. Bead-bound proteins were eluted by incubation at 65°C with elution buffer (1% SDS, 100 mM Tris, pH 8.0). Eluted proteins were precipitated with 20% TCA for 1 hr. Protein precipitates were washed three times with ice-cold acetone and then dissolved in 8 M urea, 100 mM Tris, pH 8.5, then reduced with 5 mM TCEP (tris [2-carboxyethyl] phosphine) for 20 min, and alkylated with 10 mM iodoacetamide for 15 min in the dark. Then the samples were diluted by a factor of 4 and digested overnight at 37°C with trypsin (dissolved in 2 M urea, 1 mM $CaCl_2$, 100 mM Tris, pH 8.5). Formic acid was added to the final concentration of 5% to stop digestion reaction. After digestion, the LC-MS/MS analysis was performed on an Easy-nLC II

HPLC instrument (Thermo Fisher Scientific) coupled to an orbitrap QE-HF mass spectrometer (Thermo Fisher Scientific). Peptides were loaded on a pre-column (100 μm ID, 4 cm long, packed with C18 10 μm 120 Å resin from YMC Co., Ltd) and separated on an analytical column (75 μM ID, 10 cm long, packed with Luna C18 1.8 μm 100 Å resin from Welch Materials) using an acetonitrile gradient from 0% to 28% in 100 min at a flow rate of 200 nL/min. The top 15 intense precursor ions from each full scan (resolution 60,000) were isolated for higher-energy collisional dissociation tandem mass spectrometry spectra analysis (HCD MS2; normalized collision energy 30) with a dynamic exclusion time of 60 s. Tandem mass spectrometry fragment ions were detected using orbitrap in a normal scan mode. Precursors with charge states that were unassigned or less than 2+ were excluded. The pFind software was used to identify proteins with the cutoffs of peptide FDR < 1% (*Chi et al., 2018*).

## Pull-down assay

HA-fused Atg1(593-830) was expressed in BL21 *E. coli* strain and purified using Ni-NTA beads (QIA-GEN) according to manufacturer's instructions. GST-fused Atg11(522-544), Atg11(522-544)F526A, Atg11(522-544)Y527A, and GST alone were expressed in BL21 *E. coli* strain and affinity absorbed onto Glutathione Sepharose 4 Fast Flow beads (GE Healthcare) according to manufacturer's instructions. About 2 μg of purified HA-fused Atg1(593-830) was mixed with beads containing about 1 μg Atg11 proteins in 250 μL of pull-down buffer (50 mM $NaH_2PO_4$, pH 8.0, 300 mM NaCl, 10 mM imidazole, 10% glycerol), and incubated at 4°C for 2 hr. Beads were washed three times with wash buffer (50 mM $NaH_2PO_4$, pH 8.0, 300 mM NaCl, 20 mM imidazole, 10% glycerol) and eluted with SDS-PAGE loading buffer.

## Gel-filtration analysis

MBP fused Atg11(522-583) was expressed in BL21 *E. coli* strain and purified using Ni-NTA beads according to manufacturer's instructions. After centrifugation at 10,000 g, the purified protein was applied to a Superdex 200 10/300 Increase GL column (GE Healthcare). Elution buffer was 50 mM Tris HCl, pH 7.5, 150 mM NaCl, and 0.3 mL fractions were collected at a flow rate of 0.8 mL/min. The molecular weight of MBP-Atg11(522-583) was calculated according to the known molecular weights of BSA monomer and dimer and the linear relationship between the log10 of the molecular weights and the peak elution volumes.

## Acknowledgements

This work was supported by the National Basic Research Program of China (Ministry of Science and Technology of the People's Republic of China; 973 Program, 2014CB849901 and 2014CB849801), and by grants from the Ministry of Science and Technology of the People's Republic of China and the Beijing municipal government to M-QD and L-LD.

## Additional information

### Funding

| Funder | Grant reference number | Author |
| --- | --- | --- |
| National Basic Research Program of China | 2014CB849901 | Li-Lin Du |
| National Basic Research Program of China | 2014CB849801 | Meng-Qiu Dong |

The funders had no role in study design, data collection and interpretation, or the decision to submit the work for publication.

### Author contributions

Zhao-Qian Pan, Conceptualization, Investigation, Writing - original draft, Writing - review and editing; Guang-Can Shao, Investigation; Xiao-Man Liu, Quan Chen, Methodology; Meng-Qiu Dong,

Funding acquisition, Investigation; Li-Lin Du, Conceptualization, Funding acquisition, Investigation, Writing - original draft, Writing - review and editing

### Author ORCIDs
Zhao-Qian Pan ⬩ http://orcid.org/0000-0003-0346-0259
Xiao-Man Liu ⬩ http://orcid.org/0000-0001-9968-3988
Meng-Qiu Dong ⬩ http://orcid.org/0000-0002-6094-1182
Li-Lin Du ⬩ https://orcid.org/0000-0002-1028-7397

### Decision letter and Author response
Decision letter https://doi.org/10.7554/eLife.58073.sa1
Author response https://doi.org/10.7554/eLife.58073.sa2

## Additional files

### Supplementary files
• Supplementary file 1. Fission yeast strains and plasmids used in this study.

• Transparent reporting form

### Data availability
All data generated or analysed during this study are included in the manuscript and supporting files.

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
