## [Decision Letter]

**Acceptance summary:**

A major bottleneck in the field of autophagy is the lack of mechanistic studies. This work presents an elegant mechanistic study on the activation mechanism of the key regulatory kinase in autophagy and the subsequent induction of the process.

**Decision letter after peer review:**

Thank you for submitting your article "Atg1 kinase in fission yeast is activated by Atg11-mediated dimerization and cis-autophosphorylation" for consideration by *eLife*. Your article has been reviewed by three peer reviewers, including Claudine Kraft as the Reviewing Editor and Reviewer #1, and the evaluation has been overseen by Vivek Malhotra as the Senior Editor.

The reviewers have discussed the reviews with one another and the Reviewing Editor has drafted this decision to help you prepare a revised submission.

Summary:

The manuscript by Pan et al. presents an elegant mechanistic study of the scaffold protein Atg11 in autophagy in *S. pombe*, and its role in activating the key regulatory kinase Atg1. The authors found that in *S. pombe* Atg1 kinase activation upon nitrogen starvation requires Atg11, but not the other Atg1 kinase complex subunits Atg13, Atg17 or Atg101, which are required in other organisms. Surprisingly, the lack of Atg11 can be complemented by artificial dimerization of Atg1 kinase, which suggests that the function of Atg11 in *S. pombe* is simply to bring together two Atg1 molecules.

Overall, we think that this work is an important mechanistic study addressing the activation of the key regulatory kinase in bulk autophagy. The experiments seem mostly sound. We identified four major points that need to be addressed.

Essential revisions:

1) The work describes a possibly unique situation in *S. pombe*. To broaden the significance of these findings, the situation also for selective autophagy in *S. pombe* should be addressed. Is Atg11 required for selective autophagy pathways (e.g. mitophagy) in *S. pombe*? If yes, is also dimerization required? Is the dimerizing Atg11 fragment also sufficient to complement *atg11Δ* cells in selective autophagy or is more of the Atg11 protein required?

2) In Figure 1D Atg1 kinase activity seems as high under nutrient rich conditions as under starvation. Is this indeed the case? Also, for the nutrient rich condition it seems that deletions of Atg13, Atg17 and Atg101 indeed seem to lower Atg1 autophosphorylation, whereas these deletions do not affect the starved situation.

To clarify these findings, the kinase assay should be repeated under nutrient rich and starved conditions with the wild type and the deletions also using the GST-fused peptide as a substrate. Both the presented and the new kinase assay needs quantification of at least three independent experiments to clarify these findings. The respective findings should be clearly stated and discussed in the text.

3) In Figure 3 the authors show that Atg11-546-583 is sufficient for homodimerization of Atg11. The analysis is only done with Atg11 fragments. To strengthen their conclusion, the authors should show that a) Atg11 full length indeed dimerizes and b) that Atg11-full length with a deletion of the region 546-583 (or a mutation therein) fails to dimerize and fails to promote kinase activation and autophagy.

4) Experiments should be performed in triplicates and stated as such. Especially for Figure 1D a quantification is essential. Figures 1H and 4B would also be more convincing with a quantification.

---

## [Author Response]

Essential revisions:1) The work describes a possibly unique situation in S. pombe. To broaden the significance of these findings, the situation also for selective autophagy in S. pombe should be addressed. Is Atg11 required for selective autophagy pathways (e.g. mitophagy) in S. pombe? If yes, is also dimerization required? Is the dimerizing Atg11 fragment also sufficient to complement atg11Δ cells in selective autophagy or is more of the Atg11 protein required?

We agree with the reviewers that the role of Atg11 in selective autophagy is worth addressing. ER stress-induced selective ER-phagy is the only selective macroautophagy pathway with a defined autophagy receptor in *S. pombe* (Zhao et al., 2020). Thus, we performed additional experiments to examine the role of Atg11 in this selective autophagy pathway, and found that Atg11 is required for ER stress-induced selective ER-phagy. Furthermore, we found that Atg11(522-583), the 62-amino-acid fragment sufficient for the bulk autophagy function of Atg11, can rescue the ER-phagy defect of *atg11Δ*. Atg11(522-552), a shorter fragment that cannot dimerize, failed to rescue the ER-phagy defect of *atg11Δ*. Adding a heterologous dimerization motif to Atg11(522-552) rendered it capable of rescuing the ER-phagy defect of *atg11Δ*. Thus, the dimerization ability of Atg11 is important for selective autophagy. These results are shown in Figure 3—figure supplement 2 of the revised manuscript.

2) In Figure 1D Atg1 kinase activity seems as high under nutrient rich conditions as under starvation. Is this indeed the case? Also, for the nutrient rich condition it seems that deletions of Atg13, Atg17 and Atg101 indeed seem to lower Atg1 autophosphorylation, whereas these deletions do not affect the starved situation.To clarify these findings, the kinase assay should be repeated under nutrient rich and starved conditions with the wild type and the deletions also using the GST-fused peptide as a substrate. Both the presented and the new kinase assay needs quantification of at least three independent experiments to clarify these findings. The respective findings should be clearly stated and discussed in the text.

We have performed additional kinase assays and quantitated the results, as suggested by the reviewers. Atg1 kinase activity towards itself and the non-self substrate appeared to be higher under starved conditions than under nutrient rich conditions, but the difference is not statistically significant. Atg1 kinase activity towards itself and the non-self substrate exhibited no significant reduction in mutants lacking Atg13, Atg17, or Atg101. These data are shown in Figure 1D and Figure 1—figure supplement 1A of the revised manuscript.

3) In Figure 3 the authors show that Atg11-546-583 is sufficient for homodimerization of Atg11. The analysis is only done with Atg11 fragments. To strengthen their conclusion, the authors should show that a) Atg11 full length indeed dimerizes and b) that Atg11-full length with a deletion of the region 546-583 (or a mutation therein) fails to dimerize and fails to promote kinase activation and autophagy.

We have followed the suggestions of the reviewers and performed additional experiments on wild-type full-length Atg11 and full-length Atg11 with an internal deletion of the 546-583 region. The results showed that wild-type full-length Atg11 but not Atg11(546-583)Δ can self-interact. Furthermore, Atg11(546-583)Δ cannot support Atg1 kinase activity and starvation-induced bulk autophagy. These results are shown in Figure 3—figure supplement 1 of the revised manuscript.

4) Experiments should be performed in triplicates and stated as such. Especially for Figure 1D a quantification is essential. Figures 1H and 4B would also be more convincing with a quantification.

We have performed quantifications as suggested by the reviewers, and added the quantification results to Figures 1D, H, and 4B.